# Multi-Objective Optimization of Beverage Based on Lactic Fermentation of Goat's Milk Whey and Fruit Juice Mixes by Kefir Granules

Diana Alexandra Nastar Marcillo, Valeria Olmedo Galarza, Nicolás Sebastián Pinto Mosquera, Rosario del Carmen Espín Valladares, Jimmy Núñez Pérez  and José Manuel Pais-Chanfrau *

Agroindustrial Engineering Career, FICAYA Faculty, Universidad Técnica del Norte, "17 de Julio" Avenue and Str. General José María de Córdova, Ibarra 100115, Imbabura, Ecuador
* Correspondence: jmpais@utn.edu.ec

**Abstract:** Numerous fruits are produced in Ecuador, of which about 40% are never eaten. In addition, fresh goat cheeses are in high demand. However, goat cheese generates goat milk whey with high contamination loads, and, therefore, it must be adequately treated before being discharged into ecosystems. This research aims to use a mixture of tree tomato, common strawberry juices, and goat's milk whey, to be statically fermented by milk and water kefir grains (WKG) for 48 h. For this, a dual mixture design of L-optimal response surface methodology was carried out to find the conditions that maximized all the responses evaluated (lactic-acid bacteria and yeasts concentrations and the overall acceptability assessed on a 7-point scale). Experiments were carried out in San Gabriel, Ecuador. Temperatures during the day and night were 20.2 ± 0.3 °C and 18.7 ± 0.3 °C, respectively. Three conditions were selected, where the highest response values were reached. Complementary experiments demonstrated the validity of the models. When comparing the results of the present study with similar ones carried out previously, higher values were observed in the concentration of yeasts, which seems related to the presence of the WKG. It is concluded that they could be suitable functional beverage candidates.

**Keywords:** goat milk whey valorisation; tree-tomato (*Solanum betaceum*); common strawberry (*Fragaria × ananassa*); multi-objective optimization; response surface methodology; functional beverage

## 1. Introduction

The excellent nutritional properties of goat milk are well known [1]. The composition of goat milk's bioactive components (such as calcium, protein, fat, vitamin A, and vitamin B1) is equal to or exceeds cow's milk [2], except for lower values of lactose in goat milk, making it attractive to people with partial lactose intolerance [3,4]. Additionally, fat globules are smaller than those found in cow's milk, which is believed to be associated with the greater digestibility of goat's milk [5]. Some of these nutritional properties are also preserved in its whey, the main by-product of cheese production, and one of the leading destinations for Ecuador's dairy production [6].

On the other hand, the taste of goat's milk differs from that of cow's milk and may not be attractive to some consumers of cow's milk [7,8]. The characteristic goat flavor is attributed to some distinct acids (such as caproic, caprylic, and capric acids) associated with goat milk fat [9]. These particular flavors also transfer to goat milk whey and must be considered if a functional beverage candidate that takes advantage of the nutritional properties of goat milk whey is to achieve customer preference [10]. One way to mask these unpleasant flavors is to blend goat milk whey (GMW) with fruit juices [10–12]. In addition to improving taste and flavor, fruit juices could provide beverages with vitamins and other nutrients [13].

In the Andean region of Ecuador, tree tomato (*Solanum betaceum*) and common strawberry (*Fragaria × ananassa*) juices are popular; the former being a genuinely Andean fruit [14], however the latter may have a remote origin in the Andes, and is most likely obtained from numerous crossings of original fruits from Americas and Europe fruits [15].

On the other hand, kefir is a renowned fermented milk drink from the Northern Caucasus region and is very popular in Eastern Europe and Central Asia [16–18]. Functional and nutraceutical properties are attributed to it due to the presence of lactic- and acetic acid bacteria and yeasts present in kefir grain, with probiotic properties and possibly, also, to the presence of peptides and polysaccharides, with recognized antioxidant and prebiotic characteristics [19–21].

Originally, kefir granules were adapted to use the nutrients in milk, mainly lactose [22]. Subsequently, it is possible by their common origin [17] that some of these granules, and the consortium of microorganisms that cohabit in them, were adapted to grow and propagate in various infusions, juice, and vegetable beverages, and in non-lactic juices [23–26]. The former are called milk kefir grains (MKG), and the latter are called water kefir grains (WKG) [17].

The use of milk- or water-kefir grains (MKG and WKG) have been used as starters for the fermentation processes of substrates such as milk [27,28], and whey [12], herbal teas [29], vegetable juices [30], and fruit juices [31,32]. Numerous reports indicate the health benefits of beverages fermented with these granules [17,26,33–37].

Fermented beverages with nutraceutical properties from agricultural by-products, such as goat milk whey (GMW), constitute an attractive and beneficial alternative for consumers and the environment [38].

Due to its high concentration of lactose in goat's milk whey (GMW), as occurs with its similar concentration obtained from cows, it could constitute a problem for the environment [39,40]. Whey is the most polluting and dangerous by-product of the cheese industry for the environment [41,42].

Therefore, making a fermented beverage, such as the ones obtained in the present study, which possesses probiotic and prebiotic properties, and with high levels of antioxidant capacity, could be an attractive alternative, compared to assuming the high costs that would be required for its treatment before its deposition to the environment as a high-polluted agro-industrial waste [43,44].

However, few studies have explored the simultaneous use of MKG and WKG when using more complex mixtures, such as those studied here between goat's milk whey and a mix of fruit juices. As the microbiota better adapted to the transformation of sugars other than lactose, it cohabits with the one that gave rise to it, accustomed to metabolizing lactose.

In the present work, a fermented functional beverage candidate made from goats' milk whey and a mixture of tree tomato and ordinary strawberry juice, in different proportions, was elaborated. For the fermentation, kefir granules were used, both milk (MKG) and water (WKG), in different proportions. The objective of the work was to find the most suitable combination for which a specific combined desirability function is maximized, formed by the concentrations of lactic acid bacteria (LAB), and the yeasts present in the final juice, as well as the overall-acceptability score obtained by a semi-trained tasting panel.

## 2. Material and Methods

### 2.1. Milk- and Water-Kefir Grains

Milk- and water-kefir granules from a local supplier in Quito (www.kefir.ec, accessed on 19 September 2022) were used. Milk kefir and water kefir grains were kept in whole fresh milk and brown sugar-sweetened water (adding about 125 g of commercial brown sugar in a liter of water) as substrates, respectively. Both substrates were pasteurized (at 60–80 °C for 30 min) before use and maintained at room temperature after inoculating with their kefir granules. Substrates were changed every two days.

### 2.2. Goat Milk Whey Obtention and Juice Mix Preparation

Around 15 L of fresh goat's milk from the Saanen breed were used for all experiments to obtain the fresh goat's cheese and the goat's milk whey employed in this research. It was purchased at the local market in San Gabriel, Carchi, Ecuador. Initially, the milk is filtered through a fine clean white cloth to remove any foreign matter. Then, a sample of the filtered milk was analyzed for acidity, pH, temperature, density, total solids, lactose, protein, and fat, using the Ekomilk Bond Total 40s (Füssen-Equipos para la Industria Lácteas, Aguascalientes 20000, Mexico). A CryoSmart 3600 (Qingdao Antech Scientific Co., Qingdao, China) device was used for the cryoscopy point measurement. Once it has been verified that the above parameters are within the acceptable ranges for use in cheese manufacturing, the milk is pasteurized by subjecting it to a temperature of 65 °C for 30 min to reduce the concentration of potentially pathogenic microorganisms. Next, the pasteurized goat milk is cooled to a temperature of 37–39 °C, and rennet is added to the milk. It is stirred for 1 min to dissolve well and then left to stand for about 30 min for curdling. Next, the curd is cut into small squares and shaken for 10 min to facilitate the release of whey, and it is left to rest for another 5 min. Finally, after decanting the free whey, the mass is placed inside a fine white cloth and hung above the container so that the goat's milk whey continues to drain by gravity. The canvas should not be compressed to prevent small curd lumps from falling onto the whey container. With this laboratory-scale procedure, 8.9 L of goat's milk whey (GMW) and 6.5 kg of fresh goat's milk cheese was obtained from 15 L of goat's milk.

The tree tomato and the common strawberry are two of the most consumed fruits in the markets of the Ecuadorian Andean region. The first is usually consumed in the form of juice, while the second, in addition to its consumption as juice, is traditionally consumed as fresh fruit. The acceptance of both by consumers meant that they were selected to be mixed with goat whey to obtain a fermented drink using kefir grains.

Fresh ripe fruits were used each week to prepare the juices, which had no visible damage. Subsequently, the tree tomato was peeled, and the whole fresh strawberry was used. First, 3590 g of fresh ripe fruit (2000 g of tree-tomato + 1590 g of strawberry) were mixed without water and blended in a conventional domestic blender for ~30–40 s until verifying a homogeneous mixture was obtained, reaching ~1.2 L of total volume. Subsequently, the pulp juices obtained were passed through a homemade filter to retain the larger lumps. Finally, the obtained mixed pulp juices (JMX) were stored at 4 °C until use.

All the materials in contact with the raw materials, and their subsequent preparation, were carefully washed and pasteurized (at 60–80 °C for 30 min) before being used.

### 2.3. Combined Dual-Mix L-Optimal Response Surface Design

All experiments were carried out in San Gabriel, Carchi province, Ecuador. The city is located at coordinates 0°37′ N 77°50′ W and 2860 m above sea level. The room where the static fermentations were carried out for four consecutive weeks for 48 h had temperatures of 20.2 ± 0.3 °C during the day and 18.7 ± 0.3 °C at night.

In the present design, two mixtures were combined to find the optimum of a certain desirability function, where the concentrations of lactic acid bacteria and yeast are simultaneously maximized, together with the overall acceptability of the fermented beverage. The first (Mix 1), was formed by the fraction's masses of goat's milk whey (A: GMW) and a mixture of tree tomato and strawberry juices (B: JMX); and the second (Mix 2), was formed by the mass fractions of milk kefir grains (C: MKG) and water kefir grains (D: WKG). The total volume of each experimental treatment was approximately 400 mL. Of course, in both cases, the restrictions are satisfied: A + B = 1 and C + D = 1.

For the design and analysis of the experiments, the Design-Expert version 13.0.11.0 software (Stat-Easy Inc., Minneapolis, MN 55413, USA.) was used, and due to the availability of the equipment, it was organized into several blocks (weeks), as can be seen later (Table 1).

**Table 1.** Dual mix factors and their responses in the L-optimal Response Surface Methodology. Mix 1: A: GMW and B: JMX; Mix 2: C: MKG and D: WKG.

| Block | Run | Mix 1 | | Mix 2 | | Responses | | |
|---|---|---|---|---|---|---|---|---|
| | | $(x_1)$ A GMW [a] | $(x_2)$ B JMX [b] | $(x_3)$ C MKG [c] | $(x_4)$ D WKG [d] | $y_L$ LAB [e] (CFU/mL) | $y_Y$ Yeast (CFU/mL) | $y_{Ac}$ Acceptability (-) |
| week 1 | 1 | 0.77 | 0.23 | 0.77 | 0.23 | $7.34 \times 10^6$ | $3.60 \times 10^7$ | 5.25 |
| | 2 | 0.00 | 1.00 | 0.50 | 0.50 | $9.25 \times 10^7$ | $4.04 \times 10^7$ | 5.00 |
| | 3 | 0.53 | 0.47 | 0.55 | 0.45 | $5.13 \times 10^7$ | $2.96 \times 10^7$ | 6.00 |
| | 4 | 0.93 | 0.08 | 0.26 | 0.74 | $3.94 \times 10^7$ | $3.73 \times 10^7$ | 5.50 |
| | 5 | 0.00 | 1.00 | 0.50 | 0.50 | $1.66 \times 10^7$ | $4.60 \times 10^7$ | 4.75 |
| | 6 | 0.50 | 0.50 | 0.00 | 1.00 | $2.30 \times 10^7$ | $4.28 \times 10^7$ | 6.25 |
| | 7 | 0.50 | 0.50 | 0.00 | 1.00 | $4.90 \times 10^7$ | $5.92 \times 10^7$ | 6.50 |
| | 8 | 0.00 | 1.00 | 1.00 | 0.00 | $1.80 \times 10^7$ | $3.57 \times 10^7$ | 5.75 |
| week 2 | 9 | 0.50 | 0.50 | 1.00 | 0.00 | $3.67 \times 10^8$ | $2.38 \times 10^9$ | 6.50 |
| | 10 | 1.00 | 0.00 | 0.51 | 0.49 | $1.49 \times 10^9$ | $4.97 \times 10^9$ | 4.75 |
| | 11 | 1.00 | 0.00 | 0.51 | 0.49 | $1.74 \times 10^9$ | $4.88 \times 10^9$ | 4.25 |
| | 12 | 0.55 | 0.45 | 0.46 | 0.54 | $4.85 \times 10^8$ | $2.16 \times 10^9$ | 5.75 |
| | 13 | 0.26 | 0.74 | 0.21 | 0.79 | $2.72 \times 10^8$ | $2.70 \times 10^9$ | 5.75 |
| | 14 | 1.00 | 0.00 | 0.00 | 1.00 | $7.67 \times 10^8$ | $1.91 \times 10^9$ | 5.00 |
| | 15 | 0.50 | 0.50 | 1.00 | 0.00 | $8.87 \times 10^8$ | $3.61 \times 10^9$ | 6.75 |
| | 16 | 0.26 | 0.74 | 0.21 | 0.79 | $7.50 \times 10^8$ | $1.96 \times 10^9$ | 5.25 |
| week 3 | 17 | 1.00 | 0.00 | 1.00 | 0.00 | $5.68 \times 10^8$ | $7.27 \times 10^8$ | 5.00 |
| | 18 | 0.42 | 0.58 | 0.50 | 0.50 | $8.86 \times 10^7$ | $5.09 \times 10^8$ | 6.50 |
| | 19 | 0.24 | 0.76 | 0.77 | 0.23 | $6.18 \times 10^8$ | $1.20 \times 10^9$ | 5.25 |
| | 20 | 0.66 | 0.34 | 0.22 | 0.78 | $6.49 \times 10^8$ | $5.87 \times 10^8$ | 5.50 |
| | 21 | 0.00 | 1.00 | 0.00 | 1.00 | $6.09 \times 10^8$ | $2.15 \times 10^9$ | 6.50 |
| | 22 | 0.00 | 1.00 | 0.00 | 1.00 | $5.89 \times 10^8$ | $2.08 \times 10^9$ | 6.25 |
| | 23 | 1.00 | 0.00 | 1.00 | 0.00 | $5.88 \times 10^8$ | $6.97 \times 10^8$ | 5.00 |
| week 4 | 24 | 0.71 | 0.29 | 0.00 | 1.00 | $1.50 \times 10^9$ | $3.86 \times 10^8$ | 6.50 |
| | 25 | 0.00 | 1.00 | 0.00 | 1.00 | $4.49 \times 10^8$ | $8.20 \times 10^8$ | 6.30 |
| | 26 | 0.21 | 0.79 | 0.75 | 0.25 | $3.56 \times 10^8$ | $5.69 \times 10^8$ | 6.10 |
| | 27 | 0.99 | 0.01 | 1.00 | 0.00 | $5.52 \times 10^8$ | $3.79 \times 10^8$ | 6.20 |
| | 28 | 0.52 | 0.48 | 1.00 | 0.00 | $7.58 \times 10^7$ | $4.46 \times 10^8$ | 5.70 |
| | 29 | 0.00 | 1.00 | 1.00 | 0.00 | $5.95 \times 10^7$ | $4.51 \times 10^8$ | 5.90 |
| | 30 | 0.71 | 0.29 | 0.00 | 1.00 | $1.38 \times 10^9$ | $4.25 \times 10^8$ | 6.75 |
| | 31 | 0.00 | 1.00 | 1.00 | 0.00 | $1.19 \times 10^8$ | $4.82 \times 10^8$ | 6.30 |
| | 32 | 0.52 | 0.48 | 1.00 | 0.00 | $1.16 \times 10^8$ | $3.66 \times 10^8$ | 6.30 |
| | 33 | 0.99 | 0.01 | 1.00 | 0.00 | $4.63 \times 10^8$ | $2.89 \times 10^8$ | 6.15 |
| | 34 | 0.21 | 0.79 | 0.75 | 0.25 | $3.76 \times 10^8$ | $6.18 \times 10^8$ | 5.70 |
| | 35 | 0.00 | 1.00 | 0.00 | 1.00 | $4.09 \times 10^8$ | $8.61 \times 10^8$ | 6.40 |
| | 36 | 0.71 | 0.29 | 0.00 | 1.00 | $1.47 \times 10^9$ | $4.05 \times 10^8$ | 6.80 |
| | 37 | 0.00 | 1.00 | 0.00 | 1.00 | $4.79 \times 10^8$ | $8.40 \times 10^8$ | 6.10 |
| | 38 | 0.21 | 0.79 | 0.75 | 0.25 | $4.27 \times 10^8$ | $6.38 \times 10^8$ | 6.70 |
| | 39 | 0.99 | 0.01 | 1.00 | 0.00 | $4.93 \times 10^8$ | $4.09 \times 10^8$ | 6.50 |
| | 40 | 0.52 | 0.48 | 1.00 | 0.00 | $9.54 \times 10^7$ | $4.06 \times 10^8$ | 5.60 |
| | 41 | 0.00 | 1.00 | 1.00 | 0.00 | $8.96 \times 10^7$ | $3.63 \times 10^8$ | 5.80 |

[a] GMW: Goat Milk Whey; [b] JMX: Juice Mix; [c] MKG: Milk Kefir Grains; [d] WKG: Water Kefir Grains; [e] L.A.B.: Lactic-Acid Bacteria.

*2.4. Physical-Chemical and Microbiological Determinations of the Fermented Beverages*

The tree tomato (*Solanum betaceum*) and common strawberry (*Fragaria × ananassa*) are two of the most consumed fruits in the markets of the Ecuadorian Andean region. The first is usually consumed as juice, while the second, in addition to its consumption as juice, is traditionally consumed as fresh fruit. The acceptance of both by consumers meant that they were selected to be mixed with goat whey to obtain a fermented drink using kefir grains.

Lactose concentration was determined by RP-HPLC, according to a method described elsewhere (AOAC 982.14-1983, Glucose, Fructose, Sucrose, and Maltose in Presweetened Cereals—Liquid Chromatographic Method) [45].

Crude fat and proteins were determined according to a commonly accepted (AOAC 2003.06-2006, Crude Fat in Feeds, Cereal Grains, and Forages [46]; and AOAC 2001.11-2001, Protein (crude) in animal feed, forage, grain, and oilseed [47]). At the same time, the

alcoholic degree and viscosity were measured according to the standards of the United States Pharmacopeia (USP-NF <611> Alcohol Determination) and United States standards (ASTM D 446: 2012: R2017), respectively.

The titratable acidity of the samples (as % wt. of lactic acid), a titrimetric method with 0.1 M NaOH and phenolphthalein indicator (AOAC 947.05-1947, Acidity of milk. Titrimetric method, 1947) was used [48].

Finally, 3M™ Petrifilm™ Lactic Acid Bacteria Count Plates were used for LAB counting, and 3M™ Petrifilm™ Yeast and Mold Count Plates (both from 3M Science. Applied to life™, Alexandria, MN 56308, USA) for yeast count. The populations of lactic acid bacteria (LAB) and yeasts were counted at the beginning and end of each fermentation process (48 h). All plates were incubated at 28 °C between 48–72 h for LAB plates and 72–96 h for yeast plates. The results were expressed in (CFU/mL).

### 2.5. Sensory Analysis and Overall Acceptability Score

A panel of 15 semi-trained tasters evaluated the different beverage treatments, identifying elemental odors, colors, textures, and flavors (salty, acid, bitter, and sweet) using a 7-point hedonic scale to obtain global acceptability results for each preparation [49].

The scale included the following points: 7: I like it a lot/very high; 6: I like it a little/high; 5: I neither like it nor dislike it/moderate-high; 4: I dislike it a little/medium-low; 3: I dislike/slightly; 2: I dislike a lot/very slightly; and 1: Disgusting/imperceptible [50].

The data obtained and recorded are evaluated through the non-parametric Kruskal-Wallis tests because some sensory values are non-normally distributed.

### 2.6. Antioxidant Measurement

The antioxidant capacity of the beverages obtained under optimal conditions was determined according to the 2,2 diphenyl-1-picrylhydrazyl (DPPH) assay [51] with Trolox as the standard. Both reagents were from Sigma-Aldrich (Sigma-Aldrich, St. Luis, MO, USA). Briefly, 1 mL of sample was mixed with 3 mL of 60 mM DPPH solution. Then, the samples were incubated in the dark for 30 min. Finally, the absorbance was measured at 517 nm employing a U.V./Vis spectrophotometer (Analytikjena Specord 250 plus, Analytik Jena GmbH, Konrad-Zuse-Strasse 107745 Jena, Germany) by using 80% (*v*/*v*) methanol as a control.

### 3. Results

### 3.1. Combined Dual-Mix L-Optimal Experiments

The L-optimal double-mix design of the experiments involved carrying this out in 3 blocks (weeks), after which models were obtained whose optimizations yielded optimal mixes that could not, however, be validated in complementary experiments made in the fourth week. Finally, the initial data (from the first 3 weeks) were added to those obtained in the fourth week, and with them, the new models were obtained (Table 1).

With all experimental data, the following models were obtained from the three responses evaluated:

$$Ln \ln(\hat{y}_L) = 20.26 \cdot AC + 18.67 \cdot AD + 18.31 \cdot BC + 20.02 \cdot BD - 3.14 \cdot ABC - 0.6267 \cdot ABD \\ -26.31 \cdot ABC(A - B) + 24.02 \cdot ABD(A - B) \tag{1}$$

$$\ln(\hat{y}_Y) = 19.76 \cdot AC + 19.37 \cdot AD + 19.96 \cdot BC + 20.72 \cdot BD + 0.0708 \cdot ABC + 0.8268 \cdot ABD \\ +3.06 \cdot ACD - 0.7522 \cdot BCD - 12.45 \cdot ABCD + 7.55 \cdot BCD(C - D) \tag{2}$$

$$\hat{y}_{Ac} = 5.71 \cdot AC + 4.49 \cdot AD + 6.11 \cdot BC + 6.74 \cdot BD - 0.7744 \cdot ABC + 1.26 \cdot ABD - 4.39 \cdot ACD \\ -7.58 \cdot BCD + 25.10 \cdot ABCD + 69.88 \cdot ABC(A - B) + 18.99 \cdot ABD(A - B) \\ -243.30 \cdot ABCD(A - B) - 222.52 \cdot ABCD(AC - AD - BC + BD) \tag{3}$$

The analysis of variance (ANOVA) for response models shows the significance of each term in the model (Table 2).

**Table 2.** ANOVA for dual-mix L-optimal R.S.M. on the transformed responses of LAB (upper), Yeast (middle) and Acceptability (lower).

| Source | Sum of Squares | df | Mean Square | F-Value | *p*-Value |
|---|---|---|---|---|---|
| *ANOVA for Dual-Mix Model of Transformed $\hat{y}_L$: LAB (CFU/mL)* | | | | | |
| *Block* | 49.76 | 3 | 16.59 | | |
| **Model** | 23.37 | 7 | 3.34 | 19.37 | <0.0001 |
| *Linear × Linear mixture* | 10.97 | 3 | 3.66 | 21.22 | <0.0001 |
| ABC | 2.03 | 1 | 2.03 | 11.76 | 00018 |
| ABD | 0.0507 | 1 | 0.0507 | 0.2940 | 0.5917 |
| ABC (A − B) | 9.73 | 1 | 9.73 | 56.42 | <0.0001 |
| ABD (A − B) | 7.52 | 1 | 7.52 | 43.63 | <0.0001 |
| **Residual** | 5.17 | 30 | 0.1724 | | |
| Lack of Fit | 2.11 | 11 | 0.1917 | 1.19 | 0.3563 |
| Pure Error | 3.06 | 19 | 0.1612 | | |
| **Cor. Total** | 78.30 | 40 | | | |
| *ANOVA for Dual-Mix Model of Transformed $\hat{y}_Y$: Yeast (CFU/mL)* | | | | | |
| *Block* | 78.30 | 3 | 26.10 | | |
| **Model** | 4.74 | 9 | 0.5271 | 31.42 | <0.0001 |
| *Linear × Linear mixture* | 1.98 | 3 | 0.6600 | 39.35 | <0.0001 |
| ABC | 0.0008 | 1 | 0.0008 | 0.0501 | 0.8245 |
| ABD | 0.0565 | 1 | 0.0565 | 3.37 | 0.0771 |
| ACD | 0.6926 | 1 | 0.6926 | 41.29 | <0.0001 |
| BCD | 0.0534 | 1 | 0.0534 | 3.18 | 0.0852 |
| ABCD | 0.6423 | 1 | 0.6423 | 38.29 | <0.0001 |
| BCD (C − D) | 1.08 | 1 | 1.08 | 64.24 | <0.0001 |
| **Residual** | 0.4697 | 28 | 0.0168 | | |
| Lack of Fit | 0.1241 | 9 | 0.0138 | 0.7581 | 0.6546 |
| Pure Error | 0.3456 | 19 | 0.0182 | | |
| **Cor. Total** | 83.52 | 40 | | | |
| *ANOVA for Dual-Mix Model of $\hat{y}_{Ac}$ : Acceptability (-)* | | | | | |
| *Block* | 3.84 | 3 | 1.28 | | |
| **Model** | 9.14 | 12 | 0.7617 | 5.14 | 0.0003 |
| *Linear × Linear mixture* | 1.11 | 3 | 0.3707 | 2.50 | 0.0826 |
| ABC | 0.0702 | 1 | 0.0702 | 0.4732 | 0.4979 |
| ABD | 0.1039 | 1 | 0.1039 | 0.7005 | 0.4105 |
| ACD | 1.35 | 1 | 1.35 | 9.09 | 0.0058 |
| BCD | 3.18 | 1 | 3.18 | 21.46 | <0.0001 |
| ABCD | 1.88 | 1 | 1.88 | 12.66 | 0.0015 |
| ABC (A − B) | 0.8924 | 1 | 0.8924 | 6.02 | 0.0215 |
| ABD (A − B) | 1.69 | 1 | 1.69 | 11.43 | 0.0024 |
| ABCD (A − B) | 1.17 | 1 | 1.17 | 7.89 | 0.0095 |
| ABCD (AC − AD − BC + BD) | 1.00 | 1 | 1.00 | 6.75 | 0.0155 |
| **Residual** | 3.71 | 25 | 0.1483 | | |
| Lack of Fit | 2.23 | 6 | 0.3714 | 4.77 | 0.0040 |
| Pure Error | 1.48 | 19 | 0.0778 | | |
| **Cor. Total** | 16.69 | 40 | | | |

Additionally, residuals distribution and their normality (Figure 1(a1–a3,b1–b3)) in each model, and correspondence between the values obtained by the models with the actual values, were checked (Figure 1(c1–c3)).

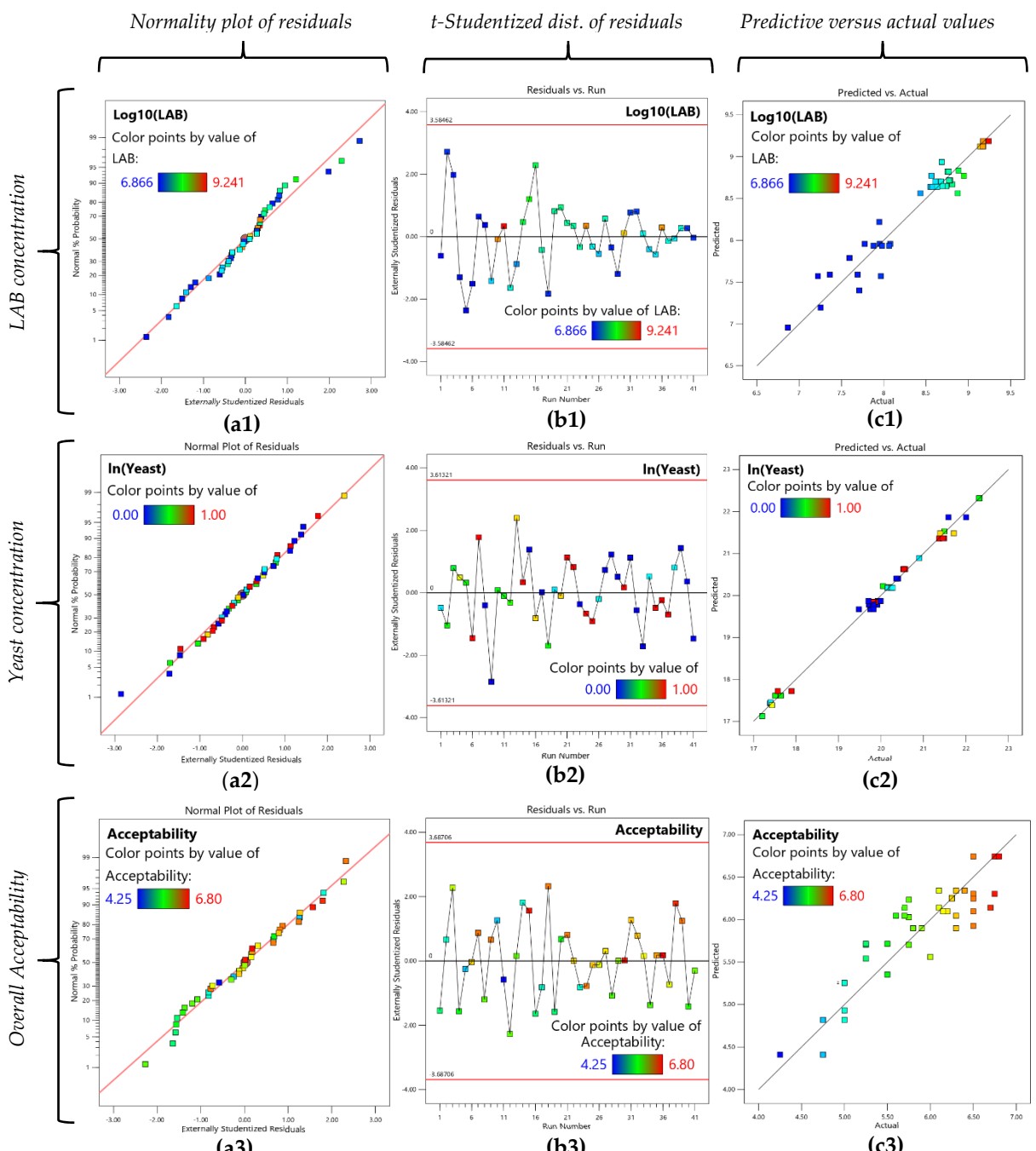

**Figure 1.** Analysis of the models of the responses for LAB concentration (upper part, **a1–c1**); yeast concentration (middle part, **a2–c2**); and overall acceptability (lower part, **a3–c3**). (**a1–a3**) Normality plot of the residuals for the models shown in Equations (1) to (3), respectively. (**b1–b3**) Student's t external distribution of the residuals. (**c1–c3**) Correspondence between the real values of the responses with the values obtained with the models shown in Equations from (1) to (3), respectively.

All the analyses showed the usefulness of the individual models and suggested they could use them to navigate on their surfaces to find the optimal values.

The contour and 3D plots of the responses show a complex relationship between them and the factors under study (Figure 2).

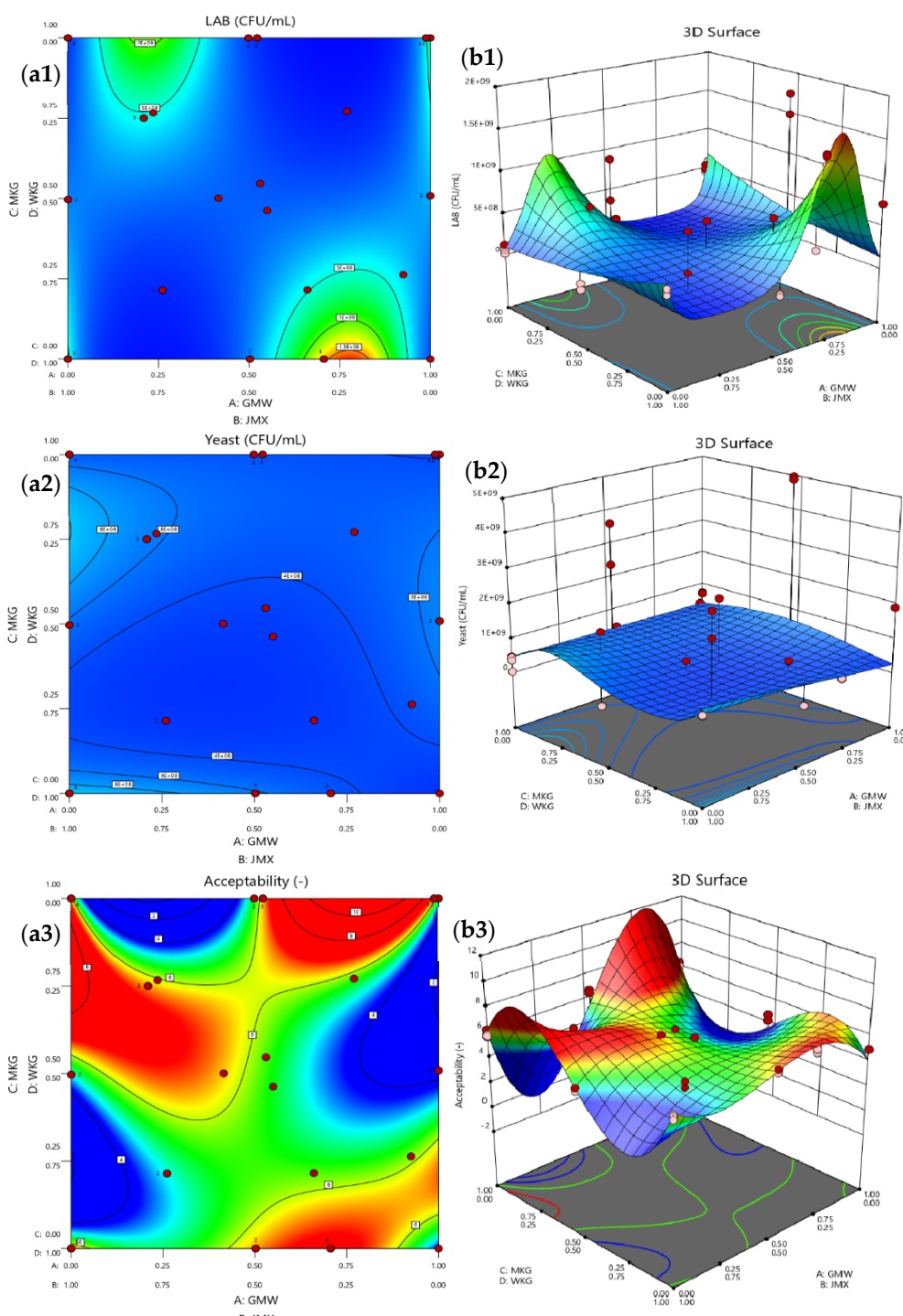

**Figure 2.** Contour (**a1**–**a3**) and 3D (**b1**–**b3**) plots for the models of the responses expressed by equations from (1) to (3) for LAB concentration ($\hat{y}_L$, **a1**,**b1**, upper), yeast concentration ($\hat{y}_Y$, **a2**,**b2**, middle), and overall acceptability ($\hat{y}_{Ac}$, **a3**,**b3**, lower), respectively.

### 3.2. Optimisation Dual-Mix L-Optimal Models

The idea of this research was to test mixtures of goat's milk whey with a juice made of tree tomato and strawberry, to reduce the characteristic flavor of the former. For this, milk kefir grains, water kefir grains, or a mixture of these, were also used, as suggested by the planning of the dual-mixture L-optimization experiment.

In addition to each model response ($\hat{y}_L$, $\hat{y}_Y$, $\hat{y}_{Ac}$) in multi-objective optimization, it could build a desirability function [52], which englobe all the responses. In this case, it will be assigned the relative importance of one response concerning the other responses. The relative importance ($r_j$, were $j = L$, $Y$, $Ac$) values range from less important: 1 (+) to very important: 5 (+++++).

The multi-objective function of desirability (D), in which the relative importance of each response concerning the rest is considered.

The highest quantity of LAB and yeasts diluted in the liquid medium of the mixture is sought, but at the same time it is needed to obtain a favorable opinion from the consumers. Therefore, three answers had to be combined. The possible combinations in which these are intertwined within the desirability D-function could be huge. Consequently, it was decided to give the three answers a similar level of importance, and that these were at their average value, obtaining eight possible optimal values, choosing the first three to carry out the complementary experiments to validate the models and the physicochemical, and nutritional characterization.

There are local maxima in each of the individual responses, which may not coincide with the maximum values of the others (Figure 2). Moreover, due to the complexity of function D, it will be challenging to conjugate the values of mixtures 1 and 2 to achieve a single optimum of this function.

In the present case, when assuming an equal importance rate for each response (each with an average value of the importance of 3 (+++)), there are three defined zones in which the extreme values of D are highlighted (Figure 3). It is precisely in these areas that we can find the top deals of D.

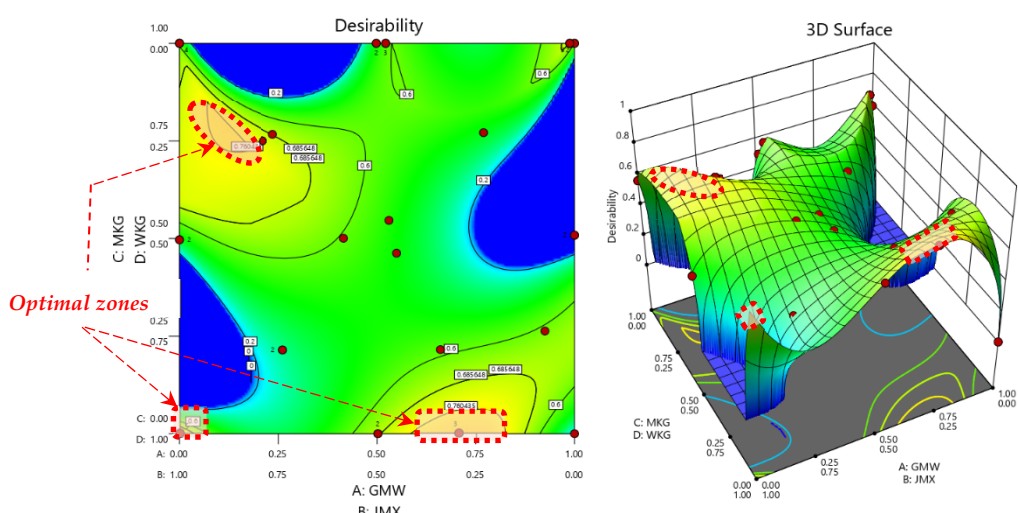

**Figure 3.** Contour (at **left**) and 3D (at **right**) plots for the desirability D-function by assuming equal and average importance (3 (+++)) of each response ($\hat{y}_L$, $\hat{y}_Y$, $\hat{y}_{Ac}$).

By optimizing the desirability function with equal levels of importance (3 (+++)) for the three responses, eight possible candidate extreme values were obtained (Table 3), the first three being those with the highest values of D (Figure 4). The first three values in Table 3 will be referred to as 'OPT1', 'OPT2', and 'OPT3', in the same order, and were used for model validation experiments, and for evaluations of their chemical-physical properties and organoleptic comparisons.

**Table 3.** Conditions of factors A-D, in mixtures 1 and 2, for which the suggested optimal values are obtained, by simultaneously maximizing the individual responses, with average and equal levels of relative importance between them.

| No. | GMW (-) | JMX (-) | MKG (-) | WKG (-) | $\hat{y}_L$ $\times 10^8$ (CFU/mL) | $\hat{y}_Y$ $\times 10^8$ (CFU/mL) | $\hat{y}_{Ac}$ (-) | D (-) |
|---|---|---|---|---|---|---|---|---|
| *1* | *0.729* | *0.271* | *0.000* | *1.000* | *15.6* | *4.4* | *6.99* | *0.8* |
| 2 | 0.000 | 1.000 | 0.000 | 1.000 | 5.4 | 10.1 | 6.61 | 0.8 |
| 3 | 0.114 | 0.886 | 0.789 | 0.211 | 4.2 | 7.9 | 6.80 | 0.8 |
| 4 | 0.000 | 1.000 | 0.672 | 0.328 | 1.7 | 9.1 | 8.01 | 0.7 |
| 5 | 0.981 | 0.019 | 1.000 | 0.000 | 3.9 | 3.9 | 6.80 | 0.7 |
| 6 | 1.000 | 0.000 | 0.229 | 0.771 | 2.0 | 4.9 | 6.52 | 0.7 |
| 7 | 0.550 | 0.450 | 0.921 | 0.079 | 0.8 | 4.8 | 6.80 | 0.6 |
| 8 | 0.537 | 0.463 | 1.000 | 0.000 | 0.8 | 4.3 | 6.80 | 0.6 |

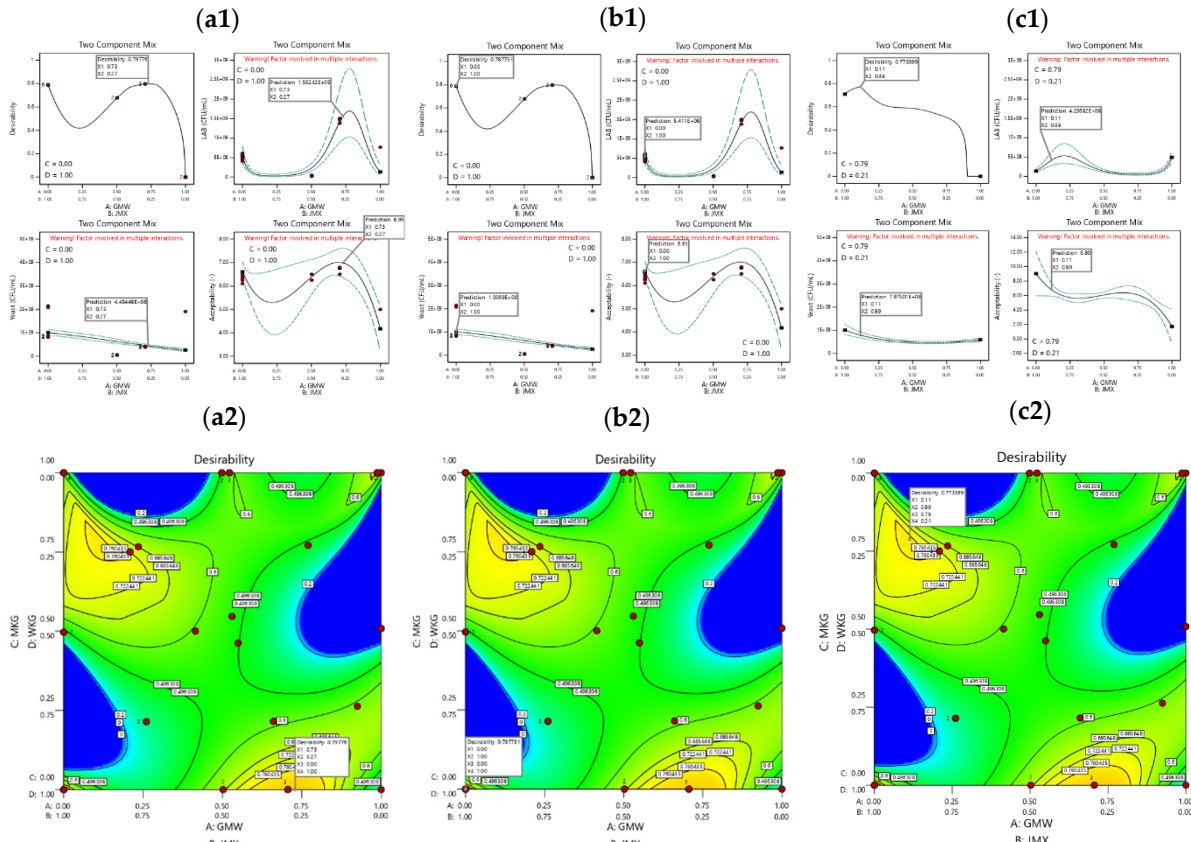

**Figure 4.** The three optimal values suggested with the highest values of desirability D-function were used in the validation experiments of the models and for the comparison of the chemical-physical properties. Upper side (**a1**–**c1**) multiple responses under optimal conditions. Lower side (**a2**–**c2**) 2D plots of desirability function. (**a1,a2**) OPT1 ($x_1$ (A) = 0.73, $x_2$ (B) = 0.27; $x_3$ (C) = 0.00, $x_4$ (D) = 1.00); (**b1,b2**) OPT2 ($x_1$ (A) = 0.00, $x_2$ (B) = 1.00; $x_3$ (C) = 0.00, $x_4$ (D) = 1.00); (**c1,c2**) OPT3 ($x_1$ (A) = 0.11, $x_2$ (B) = 0.89; $x_3$ (C) = 0.79, $x_4$ (D) = 0.21).

### 3.3. Validation of the Models

Next, five confirmatory experiments were performed under the optimal condition of factors A through D for each of the three suggested optimal values (OPT1, OPT2, and OPT3), to check the validity of the transformed models of the mixtures (Equations (1)–(3)) (Table 4).

**Table 4.** Five confirmatory experiments of the models (Equations (1)–(3)) for each optimum. Responses under optimal values compositions of factors A-D of mixes 1 and 2 fall within the acceptable values ranges ($p < 0.05$).

| Optima | Responses | Predicted Mean | Predicted Median | Std Dev | 95% P.I. Low | Data Mean | 95% P.I. High |
|---|---|---|---|---|---|---|---|
| OPT1 | $y_L \times 10^8$ | 15.63 | 14.34 | 6.78 | 7.76 | 13.08 | 26.51 |
| | $y_Y \times 10^8$ | 4.43 | 4.40 | 0.58 | 3.72 | 4.54 | 5.20 |
| | $y_{Ac}$ | 6.99 | 6.99 | 0.38 | 6.27 | 6.62 | 7.71 |
| OPT2 | $y_L \times 10^8$ | 5.41 | 4.96 | 2.35 | 2.88 | 5.07 | 8.57 |
| | $y_Y \times 10^8$ | 10.10 | 9.98 | 8.38 | 8.38 | 9.79 | 11.90 |
| | $y_{Ac}$ | 6.61 | 6.61 | 6.05 | 6.05 | 6.06 | 7.17 |
| OPT3 | $y_L \times 10^8$ | 4.20 | 3.86 | 1.82 | 2.22 | 3.95 | 3.95 |
| | $y_Y \times 10^8$ | 7.88 | 7.81 | 1.02 | 6.38 | 6.52 | 9.57 |
| | $y_{Ac}$ | 6.80 | 6.80 | 0.38 | 5.84 | 5.90 | 7.76 |

The confirmatory experiments showed a good correspondence with the models ($p < 0.05$), demonstrating their validity.

Finally, samples of the functional beverages produced under the expected conditions of the optimal OPT1, OPT2, and OPT3, as well as of the raw materials used for their preparation, the goat's milk whey and the mixture of juices, were analyzed to determine a set of physical-chemical, microbial, and nutritional properties, of the functional beverage candidates (OPT1, OPT2, and OPT3) (Figure 5).

The protein contents increased in the three optimal beverages obtained concerning the contents of the raw materials used to obtain them. These values were 9.30, 1.80, and 4.20, for the optimal 'OPT1', 'OPT2', and 'OPT3', respectively (Figure 5a).

In 'OPT1', the protein concentration was increased by 29%, while in 'OPT3' and 'OPT2', the proteins were increased by 184% and 329%, respectively. This increase is perhaps associated with more significant amounts of JMX, related to the presence of free sugars in these, which yeasts could metabolize. The optimal 'OPT2' and 'OPT3' also report higher values of yeasts (Figure 5c).

A decrease of ~12% of fat is observed in the optimal 'OPT1' and 'OPT3', and as expected, it was not observed in 'OPT2'. With regards to the presence of fat concerning the starting material, a decrease of 19% was observed in 'OPT1'. In contrast, the fat presence increased in the optimal 'OPT3' and 'OPT1', which seems related to the highest concentrations of yeasts in these preparations (Figure 5a).

For acidity, an increase is observed for 'OPT1' from the initial weighted value of the starting materials of 47%. A decrease in 63% and 75% values is kept for the 'OPT3' and 'OPT2', respectively. The first could be associated with higher LAB levels in the 'OPT1', which led to higher lactic acid values, whilst the last could be related to their most elevated titers of yeast (Figure 5a).

Ethanol levels, as expected, are higher in the optimal 'OPT2' and 'OPT3' due to higher yeast titers. At the same time, the viscosity was significantly higher in 'OPT1', probably due to the more significant presence of LAB and, therefore, the possible presence of high molecular weight exopolysaccharides that would increase viscosity (Figure 5b).

pH and °Brix were determined at the beginning and end, observing a drop of both pH and °Brix for all treatments (result not shown). This fact indicates that in all treatments, the microbial population grows and has been observed by other researchers [32,53,54]. Furthermore, there is a positive Pearson correlation between the relative decrease in pH and °Brix with the values reached for LAB ($r = 0.81$) and yeast ($r = 0.66$) growth.

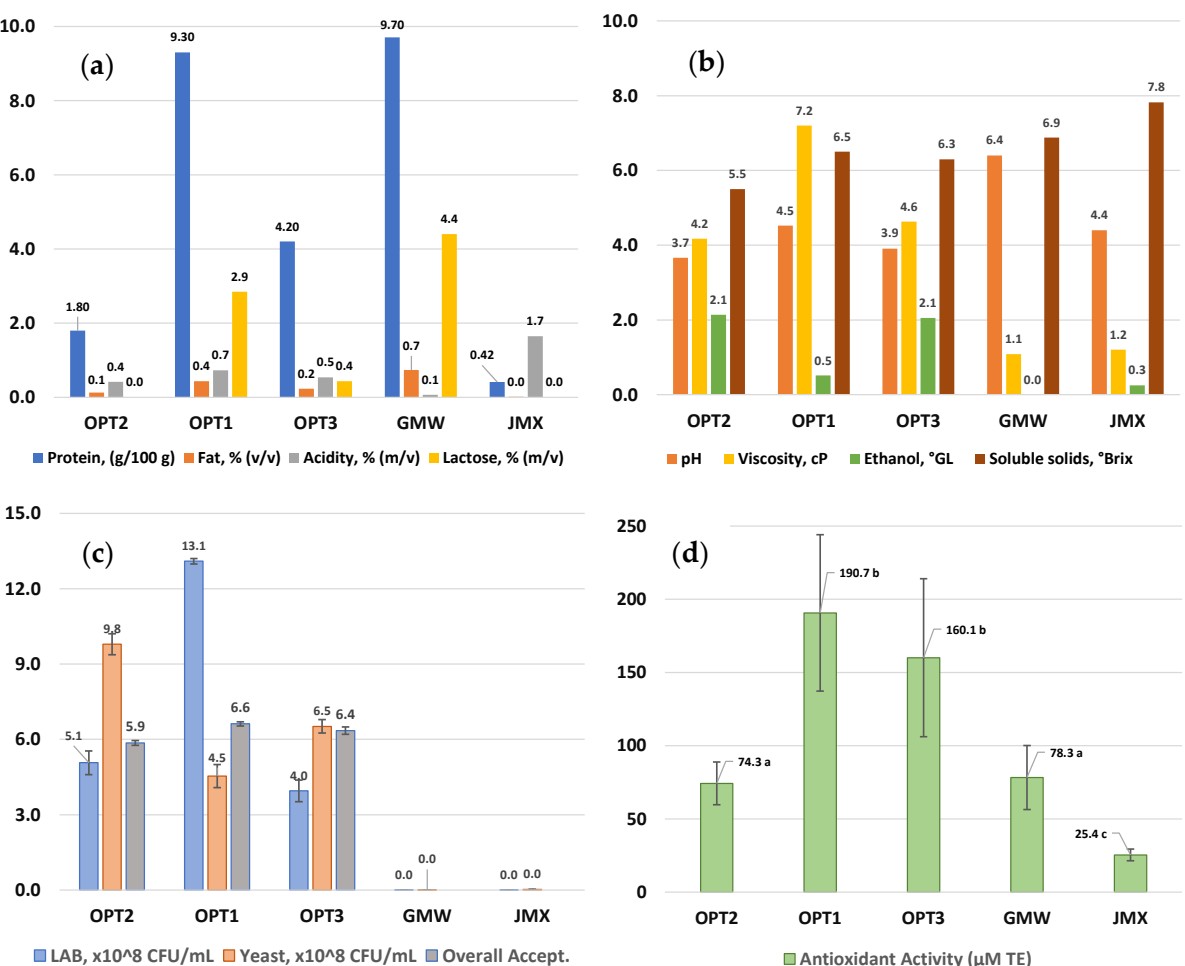

**Figure 5.** Values of the physical-chemical, nutritional, and microbiological properties, of the three candidates for functional fermented beverages produced under optimal conditions OPT1, OPT2, and OPT3, and the raw materials used in their preparation (GMW and JMX). (**a**) Nutritional properties (n = 1) protein content (g/100 g), fat (% *v/v*), lactose (% *m/v*) and acidity (% m/m); (**b**) Chemical-physical properties (n = 1) pH, viscosity (cP) and ethanol content (°GL) and soluble solids (°Brix); (**c**) Probiotic and global acceptability properties (n = 5) LAB and yeast counts (CFU/mL) and assessment of global acceptability (-); and (**d**) Antioxidant Activity (n = 3), expressed in μM of Trolox equivalent (TE). Different letters represent significant statistical differences (α < 0.05).

The levels of LAB and yeasts in the three treatments with the highest value of the D-function (D = 0.8), that have been called 'OPT1', 'OPT2' and 'OPT3', have responses significantly higher than those of the raw materials used (GMW and JMX) (Figure 5c).

Regarding the antioxidant capacity, a significant increase in its values was observed in the optimal preparations of between about two times (for 'OPT1' and 'OPT2') and four times (for 'OPT3') the values that the starting media had (GMW and/or JMX) used (Figure 5d).

Therefore, the contributions of these probiotic microorganisms, and the antioxidant capacity, can only come from the WKG and MKG used in its preparation.

Antioxidant capacity values are significantly higher for the optimal 'OPT1' and 'OPT3' that presented higher amounts of goat's whey (GMW) at the beginning. Therefore, it seems that the antioxidant capacity is related to the possible presence of small peptides in the milk and, possibly, in the goat whey.

Finally, it was interesting to observe that the antioxidant capacity values of 'OPT2', without the presence of GMW, were similar to those obtained in GMW alone, which can be

explained by the presence of water kefir grains and their activities over the components of the juice mix (JMX) during fermentation. Therefore, further studies should be conducted to understand better how kefir grains can increase the antioxidant capacity of biopreparations, even without milk and whey.

## 4. Discussion

Due to the practical benefits of kefir, numerous researchers are conducting studies to use milk, or water kefir grains, for nutraceutical beverages. However, their simultaneous use has not been evaluated so far.

In the present work, it is suggested to obtain candidates for functional nutraceutical beverages from agricultural and agro-industrial wastes through mathematical models obtained from an L-optimal dual-mixture design of experiments.

In two of the three optimal values ('OPT1' and 'OPT2'), only WKG were used, even though 'OPT1' has a higher volumetric fraction of GMW than the JMX. This fact suggests that, although WKG were adapted to use carbon sources other than lactose, the LABs present inside the WKG can use lactose efficiently and re-adapt quickly to its presence in the media.

The three optimal examples obtained, for their part, far exceed the values for LAB ($>10^7$ CFU/mL) and yeasts ($>10^4$ CFU/mL) (Table 4) established by the Ecuadorian (NTE INEN 2395:2011, NTE INEN 2608:2012, and NTE INEN 2609:2012) and international standards (CODEX 243:2018) for fermented cow's milk and whey that have been used here only as reference [55].

There is, however, no commonly accepted standard that includes the desirable characteristics and the acceptable values that this type of biopreparations should contain. Therefore, detailed studies must be conducted in the future to establish the characteristics and conditions these functional or nutraceutical beverages must meet to request their registration with national and international food regulatory agencies.

Other researchers have also used mixtures of cow's milk whey with juices to improve their fermentation products' nutritional and sensory properties with kefir grains [32,56–59].

In a study where pomegranate juice was mixed with whey, it was fermented with milk kefir granules for 32 h at two temperature levels (19 and 25 °C) and two inoculum levels (5 and 8% *m/v*), and satisfactory results were achieved [58]. It was shown that part of the microbial populations of the granules was transferred to the liquid medium. The best results were at 25 °C and 8% *m/v*, with LAB and yeast values of $1.58 \times 10^8$ and $\sim 2.00 \times 10^5$ CFU/mL, respectively [58].

In another example, a functional fermented beverage was obtained using whey cheese enriched with myrtle (*Myrtus communis*) juice. As a result, values of $3.40 \times 10^8$ CFU/mL for LAB, $1.62 \times 10^6$ CFU/mL for yeasts and global acceptability (on a 9-point scale) of $5.41 \pm 0.12$, under optimal conditions using a central composite design in the response surface methodology, were obtained [32].

Surprisingly, in two of the combinations ('OPT1' and 'OPT2'), where a better behavior was observed (with values of D = 0.8), there were those where only the WKG was used, even though in one of them ('OPT1'), the substrate mixture used contained >70% GMW. The LAB values achieved were 8-fold higher than those reported by Sabokbar and Khodaiyan [58] and 3.8-fold higher than those reported by M'Hir et al. [32], while for yeasts, they were 2270-fold higher than those reported by Sabokbar and Khodaiyan [58], and 280-fold higher than those reported by M'Hir et al. [32].

Therefore, it can be stated that the LAB present in the WKG is capable of successfully metabolizing the lactose present in the medium that comes from the GMW. Furthermore, it was interesting to note that the WKG re-adapted rapidly (less than 48 h) to the conditions of this mixture.

The 'OPT2' constitutes a typical fermentation with only water kefir grains (WKG) used as the only raw material, the juice of mixed fruits (JMX only). The LAB values reached were the same order of magnitude as those reported by Sabokbar and Khodaiyan [58] and

M'Hir et al. [32]. On the other hand, the values obtained for the yeasts were notably higher than other studies, reaching values 4800-fold higher than those reported by Sabokbar and Khodaiyan [58] and 600-fold higher than those of M'Hir et al. [32].

Finally, the 'OPT3' is the only condition that uses mixtures of substrates and granules simultaneously. In a certain way, the values of LAB, yeasts, and overall acceptability, reached intermediate values between the values reached by the previous optimal ones (Table 4). Regarding the studies reported by other researchers, their LAB values were similar, although those of yeasts were notably higher [32,58].

It was confirmed that yeast concentrations increased markedly, probably due to the presence of water kefir grains, both when they were used alone (as occurs in the optimal 'OPT1' and 'OPT2') or in their mixture with the milk kefir grains (as occurs with the 'OPT3'). The presence of certain specific yeasts, it seems, gives beverages a more pleasant taste, and they can coexist symbiotically with lactic acid bacteria (LAB) and acetic acid bacteria (AAB), as shown in a recent study, where they reached levels of more than 8 log CFU per ml [60], similar to the values obtained in this work.

The optimal values achieved in this study could be aimed at a different group of consumers. Due to many proteins, probiotics, and antioxidant agents, 'OPT1' is the functional beverage candidate that could be targeted at the largest group of consumers. The functional drink candidate 'OPT2', for its part, as it does not contain lactose, could be aimed at people who are totally or partially intolerant to this disaccharide. Finally, the 'OPT3' seems to be at an intermediate point between the previous two, and could also be aimed at a broad sector of consumers.

## 5. Conclusions

In the present study, all these elements were integrated to obtain candidate nutraceutical beverages to be produced commercially. For this, a double-mixed L-optimal response surface design was carried out, where several responses were integrated into a specific desirability function, which was maximized. The optimum values obtained in these studies were compared with those of other authors, noting a certain over-growth of the yeasts, and where acceptable values of the overall acceptability of the bio-preparations were reached. Complementary studies could be performed to know the conditions and the allowed storage time. In addition, the findings of this study could be corroborated on larger scales, where it can be subjected to the taste criteria of a more significant number of consumers.

**Author Contributions:** Conceptualization, J.M.P.-C.; methodology, D.A.N.M. and J.M.P.-C.; software, J.M.P.-C.; validation, D.A.N.M., V.O.G., and R.d.C.E.V.; formal analysis, D.A.N.M. and V.O.G.; investigation, D.A.N.M., V.O.G., N.S.P.M., and J.N.P.; resources, D.A.N.M. and R.d.C.E.V.; data curation, D.A.N.M., V.O.G., N.S.P.M., and J.N.P.; writing—original draft preparation, D.A.N.M. and J.M.P.-C.; writing—review and editing, J.M.P.-C.; visualization, D.A.N.M. and J.M.P.-C.; supervision, J.M.P.-C.; project administration, D.A.N.M. and R.d.C.E.V.; funding acquisition, D.A.N.M. and R.d.C.E.V. All authors have read and agreed to the published version of the manuscript.

**Funding:** This research received no external funding.

**Institutional Review Board Statement:** Not applicable.

**Informed Consent Statement:** Not applicable.

**Data Availability Statement:** Not applicable.

**Acknowledgments:** The authors of the work wish to express their gratitude to the dean, Marcelo Cevallos, of the FICAYA, for the support given to this research.

**Conflicts of Interest:** The authors declare no conflict of interest. All authors' approved data is accurate and agreed to post in the Journal.

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
