# Peer review of "Multi-Objective Optimization of Beverage Based on Lactic Fermentation of Goat’s Milk Whey and Fruit Juice Mixes by Kefir Granules"

_fermentation, doi:10.3390/fermentation8100500_

Round 1

Reviewer 1 Report

The manuscript entitled " Multi-objective optimization of beverage based on lactic fermentation of goat's milk whey and fruit juice mixes by kefir  granules"

In the present work, a fermented functional beverage candidate made from goat's  milk whey and a mixture of tree tomato and ordinary strawberry juice, in different proportions,  was  elaborated. 

For  the  fermentation,  kefir  granules  were  used,  both  milk  (MKG) and water (WKG), in different proportions.

There are many points on which I think authors should provide more explanatory answers.

We believe that the authors failed to clearly describe the structure of the experiment. The experimental approach is not distinguished by its completeness.

There are significant deficiencies in the composition and properties of the basic raw materials used in the study.

The authors have not specified what they mean as " milk whey". They mention the phrase "Goat milk whey, without explaining its origin, (as the of cheese byproduct?  from whey strained yogurt or other origin?).

Materials and methods

The design of the experimental process and the parameters examined are not satisfactory to answer key questions.

More specific:

 lin 93-98  

2.1. Milk- and water-kefir grains   

......... Milk- and water-kefir grain from a local were  used.  Milk kefir  and  water  kefir  grains  were  kept  in  fresh  milk ?? and  brown  sugar sweetened  water (concentration???),  respectively.  Both  substrates 

room temperature and changed every two days and then???. 

lin 105-111

2.2. Juice mix preparation 

………   the juices obtained were passed through a homemade filter to retain the larger lumps . The juices obtained were stored at 4°C until use  (??? How many hours?  hygienic conditions ???).  

lin 117-124

2.3. Combined dual-mix L-optimal response surface design 

A Schematic experimental procedure to illustrate the steps involved in the study is necessary.

 lin 136-144

2.4. Physical-chemical and microbiological determinations of the fermented beverages

We believe that the description of the microbiological and physicochemical methods should be more specific

For example

We consider that more information should be provided for the microbiological analysis, as the methods currently mentioned are general for food.

Methods such as (i.e. Τotal mesophilic flora (TMF) was quantified on plate count agar (PCA), incubated at 30 °C for 72 h; LAB counts were quantified on De Man, Rogosa, and Sharpe (MRS) agar plates; yeast counts were quantified on Potato Dextrose Agar (PDA) plates, which were incubated at 30 ± 1 °C for 48 h. etc) could be used.

We consider that the evaluation and determination acetic  acid bacteria  is necessary.

We also consider that the determination of the presence of pathogenic microorganisms is necessary to be determined.

The results and discussion do not help the reader to draw clear conclusions.

This manuscript is not considered suitable for furthering our knowledge in this topic;.

My general impression is that the manuscript at its current state is weak, with substantial limitations that make it difficult to be published in the “fermentation ” Journal.

Author Response

The manuscript entitled " Multi-objective optimization of beverage based on lactic fermentation of goat's milk whey and fruit juice mixes by kefir  granules"

In the present work, a fermented functional beverage candidate made from goat's  milk whey and a mixture of tree tomato and ordinary strawberry juice, in different proportions,  was  elaborated. 

For  the  fermentation,  kefir  granules  were  used,  both  milk  (MKG) and water (WKG), in different proportions.

There are many points on which I think authors should provide more explanatory answers.

Thank you very much for the time you devoted to this thorough review of our manuscript. Next, we will try to respond to your remarks, questions, and doubts as best as possible.

 We believe that the authors failed to clearly describe the structure of the experiment. The experimental approach is not distinguished by its completeness.

In the study, an attempt was made to obtain a functional drink from the use of goat's milk whey, obtained from the manufacture of fresh goat cheese, which is widely accepted in the community. Goat's milk has a characteristic smell and taste, which is "transferred" to goat's milk whey. For certain people, more accustomed to cow's milk, the smell and taste of goat's milk and, therefore, the whey of the same name is not very pleasant. For this reason, it was decided to test mixtures of goat's milk whey (GMW), from the manufacture of fresh cheeses from goat's milk, with fruit juices. Among the latter, the tree tomato and the common strawberry are two of the most accepted and consumed. Kefir grains were used to transform the "raw material' and produce the bioactive compounds. There are two types of kefir grains, milk kefir grains (MKG) and water kefir grains (WKG). The first is perfectly adapted to grow and develop in milk and whey. The second is probably better adapted to media such as infusions, juices, and other non-dairy beverages.

As a novelty, this work has used mixtures of milk and water kefir granules in the fermentation of goat's milk whey as a mixture of juices in isolation or forming different mixtures.

Additionally, it is also desired to satisfy consumers for obvious business reasons. It was decided to establish overall acceptability as one of the answers, where sensory factors measured on a 7-point hedonic scale obtained by a semi-trained panel of 15 experts were considered.

A two-mixed L-optimal response surface design was run to achieve these goals. The mixtures that were tested were: Mixture 1: Goat's milk whey (GMW) and Pulp of a mix of tree tomato and common strawberry juices (JMX); and Blend 2: Milk Kefir Grains (MKG) and Water Kefir Grains (WKG). The DoE was focused on optimising the conditions that maximise the response variables of the study: the final concentrations of LAB and yeast and the overall acceptability. The first two are measurable objective variables, while the latter is subjective and depends on the sensory perception of a small number of people (15).

The Design-Expert version 13 program (the last version) was organised into three blocks (each block to be executed in a different week).

Statistical optimisation is a widely used tool in experimental work, and it allows finding valid answers to very complex systems and problems, such as the one presented here.

Once the experiments were executed in the random order proposed by the program, the corresponding statistical models were obtained, whose quality was verified by means of the analysis of variance and the analysis of the residuals got.

With these models, it was decided to find the condition or conditions that would allow simultaneously finding the best values ​​of the responses evaluated (the highest LAB and yeast concentrations in the beverage and the highest values ​​of general acceptability, simultaneously). Among eight possibilities, it was decided to choose the three that presented the highest values ​​of the desirability function (D-function), which were called 'OPT1', 'OPT2', and 'OPT3'.

These last variants were validated through 5 additional experiments that corroborated the validity of the obtained models.

In addition, other physical-chemical properties were measured, such as the content of protein, fat, acidity, etc., as well as the antioxidant capacity, obtaining, in all cases, values ​​higher than those detected in the starting raw materials.

Thus, the experiments presented in this study were carried out.

There are significant deficiencies in the composition and properties of the basic raw materials used in the study.

The authors have not specified what they mean as " milk whey". They mention the phrase "Goat milk whey, without explaining its origin, (as the of cheese byproduct?  from whey strained yogurt or other origin?).

Thank you very much for your observations. All these aspects related to obtaining goat's milk whey, as a by-product of the production of fresh goat cheese, were added to the new version of the manuscript.

Section "2.2. Juice mix preparation" was replaced for section "2.2. Goat milk whey obtention and juice mix preparation", where all aspects related to obtaining goat's milk whey were added

(lines 101-120).

Around fifteen litres of fresh goat milk of the Saanen breed were used for all experiments to obtain the fresh goat's cheese and the goat's milk whey employed in this research. It was purchased at the local market in San Gabriel, Carchi, Ecuador. Initially, the milk is filtered through a fine clean white cloth to remove any foreign matter. Afterwards, a sample of the filtered milk was analysed for acidity, pH, temperature, density, total solids, lactose, protein, and fat using the Ekomilk Bond Total 40s (Füssen-Equipos para la Industria Lácteas, Aguascalientes 20000, Mexico). A CryoSmart 3600 (Qingdao Antech Scientific Co., Qingdao, China) device was used for the cryoscopy point measurement. Once it has been verified that the above parameters are within the acceptable ranges for use in cheese manufacturing, the milk is pasteurised by subjecting it to a temperature of 65°C for 30 min to reduce the concentration of potentially pathogenic microorganisms. Next, the pasteurised goat milk is cooled to a temperature of 37-39°C, and the rennet is added to the milk. It is stirred for one minute to dissolve well and then left to stand for about 30 min for curdling. Next, the curd is cut into small squares and shaken for 10 min to facilitate the release of the whey, and finally, it is left to rest for another 5 min. Finally, after decanting the free whey, the mass is placed inside a fine white cloth and hung above the container so that the goat's milk whey continues to drain by gravity. The canvas should not be compressed to prevent small curd lumps from falling onto the whey container. With this laboratory-scale procedure, 8.9 L of goat's milk whey (GMW) and 6.5 kg of fresh goat's milk cheese was obtained from 15 litres of goat's milk.

Materials and methods

The design of the experimental process and the parameters examined are not satisfactory to answer key questions.

More specific:

  lin 93-98  

2.1. Milk- and water-kefir grains   

......... Milk- and water-kefir grain from a local were  used.  Milk kefir  and  water  kefir  grains  were  kept  in  fresh  milk ?? and  brown  sugar sweetened  water (concentration???),  respectively.  Both  substrates 

room temperature and changed every two days and then???. 

This section was rewritten as follows (lines 94-99):

2.1. Milk- and water-kefir grains   

Milk- and water-kefir granules from a local supplier in Quito (www.kefir.ec) were used. Milk kefir and water kefir grains were kept in whole fresh milk and brown sugar-sweetened water (adding about 125 g of commercial brown sugar in a litre of water) as substrates, respectively. Both substrates were pasteurised (at 60-80°C for 30 min) before use and maintained at room temperature after inoculating with their kefir granules. Substrates were changed every two days.

lin 105-111

2.2. Juice mix preparation 

………   the juices obtained were passed through a homemade filter to retain the larger lumps . The juices obtained were stored at 4°C until use  (??? How many hours?  hygienic conditions ???).  

lin 117-124

This section was rewritten as follows (lines 100-134):

2.2. Goat milk whey obtention and juice mix preparation

Around fifteen litres of fresh goat milk of the Saanen breed were used for all experiments to obtain the fresh goat's cheese and the goat's milk whey employed in this research. It was purchased at the local market in San Gabriel, Carchi, Ecuador. Initially, the milk is filtered through a fine clean white cloth to remove any foreign matter. Afterwards, a sample of the filtered milk was analysed for acidity, pH, temperature, density, total solids, lactose, protein, and fat using the Ekomilk Bond Total 40s (Füssen-Equipos para la Industria Lácteas, Aguascalientes 20000, Mexico). A CryoSmart 3600 (Qingdao Antech Scientific Co., Qingdao, China) device was used for the cryoscopy point measurement. Once it has been verified that the above parameters are within the acceptable ranges for use in cheese manufacturing, the milk is pasteurised by subjecting it to a temperature of 65°C for 30 min to reduce the concentration of potentially pathogenic microorganisms. Next, the pasteurised goat milk is cooled to a temperature of 37-39°C, and the rennet is added to the milk. It is stirred for one minute to dissolve well and then left to stand for about 30 min for curdling. Next, the curd is cut into small squares and shaken for 10 min to facilitate the release of the whey, and finally, it is left to rest for another 5 min. Finally, after decanting the free whey, the mass is placed inside a fine white cloth and hung above the container so that the goat's milk whey continues to drain by gravity. The canvas should not be compressed to prevent small curd lumps from falling onto the whey container. With this laboratory-scale procedure, 8.9 L of goat's milk whey (GMW) and 6.5 kg of fresh goat's milk cheese was obtained from 15 litres of goat's milk.

The tree tomato and the common strawberry are two of the most consumed fruits in the markets of the Ecuadorian Andean region. The first is usually consumed in the form of juice, while the second, in addition to its consumption as juice, is traditionally consumed as fresh fruit. The acceptance of both by consumers meant that they were selected to be mixed with goat whey to obtain a fermented drink using kefir grains.

Fresh ripe fruits were used each week to prepare the juices, which had no visible damage. Subsequently, the tree tomato was peeled, and the whole fresh strawberry was used. First, 3590 g of fresh ripe fruit (2000 g of tree-tomato + 1590 g of strawberry) were mixed without water and blended in a conventional domestic blender for ~30-40 s until verifying a homogeneous mixture was obtained, reaching ~1.2 L of total volume. Subsequently, the pulp juices obtained were passed through a homemade filter to retain the larger lumps. Finally, the obtained mixed pulp juices (JMX) were stored at 4°C until use.

All the materials in contact with the raw materials and their subsequent preparation were carefully washed and pasteurised (at 60-80°C for 30 min) before being used.

2.3. Combined dual-mix L-optimal response surface design 

A Schematic experimental procedure to illustrate the steps involved in the study is necessary.

 lin 136-144

Possibly, the requested scheme is the selection of the experimental points of the design of double-mixed response surfaces of L-optimization. The figure shown shows the positions of the 41 experimental points studied where the three responses of the present study were evaluated (LAB and yeast concentrations and overall acceptability).

< FIGURE >

The points spanned the entire design space and were “suggested” by the Design-Expert software for their rotability and orthogonality.

However, because these same points are repeated, with their respective responses in Table 1, it was decided not to redound the information and not include it as part of the manuscript.

2.4. Physical-chemical and microbiological determinations of the fermented beverages

We believe that the description of the microbiological and physicochemical methods should be more specific

For example

We consider that more information should be provided for the microbiological analysis, as the methods currently mentioned are general for food.

Methods such as (i.e. Τotal mesophilic flora (TMF) was quantified on plate count agar (PCA), incubated at 30 °C for 72 h; LAB counts were quantified on De Man, Rogosa, and Sharpe (MRS) agar plates; yeast counts were quantified on Potato Dextrose Agar (PDA) plates, which were incubated at 30 ± 1 °C for 48 h. etc) could be used.

Thank you very much for your suggestions. They will be considered in future investigations.

We consider that the evaluation and determination acetic acid bacteria is necessary.

Acetic acid bacteria are a smaller fraction than LAB in the kefir grain consortia, and their presence represents around 10-100 times less than LAB in the consortium of kefir grains. However, as noted elsewhere, their presence could be significant in the modulation and behaviour of the entire microorganism consortium.

All these suggestions will be considered in future research.

We also consider that the determination of the presence of pathogenic microorganisms is necessary to be determined.

Because, as part of the study, the sensory evaluation of the fermented beverages was expected by a semi-trained panel of 15 people, work was carried out rigorously in compliance with good practices for preparing food and drinks.

All the fruits were carefully chosen, and the goat's milk was freshly milked and came from a healthy herd. Additionally, all the material used in elaborating the GMW and JMX was washed and pasteurised. Thus, the possibility of contamination with pathogenic, adventitious, or potentially dangerous agents is relatively low.

No discomfort was reported on the day of the sensory tests, nor in subsequent ones, among any of the participating panellists.

However, if these results are scaled up, it will be pertinent to carry out complementary studies to prevent the biological risk of contamination by pathogenic microorganisms.

The results and discussion do not help the reader to draw clear conclusions.

An attempt was made to discuss the elements of the research that were found to be most interesting. First, the unusually high values of yeasts population that were obtained were always associated with the presence of water kefir grains, and second, the ability of the water kefir grains to accommodate and utilise the goat's milk whey.

The conclusions of the work were shortened for the original manuscript.

This manuscript is not considered suitable for furthering our knowledge in this topic;.

My general impression is that the manuscript at its current state is weak, with substantial limitations that make it difficult to be published in the “fermentation” Journal.

Reviewer 2 Report

Generally, the manuscript "Multi-objective optimisation of beverage based on lactic fermentation of goat's milk whey and fruit juice mixes by kefir granules" is within the scope of the journal. The manuscript (MS) contains some innovations which may be useful for the dairy industry. The MS writing quality is good and well organized, and the data is adequately presented for most parts. However, some sections need to be revised.

1. Introduction

1.1 Line 33: Two lost letters on a phrase (ng)

2. Material and Methods

2.1 Line 93: fix the word Materials for Material

2.2 Line 136-147: provide more details about physical-chemicals analysis

2.3 What is the statistical analysis used in this study? And the software? Please describe them.

3. Results 

3.1 Line 195-197: The words that indicate the figures need to be patronized in accordance with the legend of the figure. For example: If the term 1a stays in the text, the same term need describe in the legend of the figures.

4. Conclusions

4.5 Conclusions it's extensive. This topic can be reduced, focusing mainly on results obtained, and adding information about DPPH results. 

5. And please, respond to the following question: what is the real relevance of DPPH analysis for this study? In all MS this theme is not well explored.

Author Response

Generally, the manuscript "Multi-objective optimisation of beverage based on lactic fermentation of goat's milk whey and fruit juice mixes by kefir granules" is within the scope of the journal. The manuscript (MS) contains some innovations which may be useful for the dairy industry. The MS writing quality is good and well organized, and the data is adequately presented for most parts. However, some sections need to be revised.

Thank you very much for the thorough review of our manuscript. Next, I will answer each of the remarks and answer your questions and suggestions.

1. Introduction

1.1 Line 33: Two lost letters on a phrase (ng)

That was already corrected in the manuscript

2. Material and Methods

2.1 Line 93: fix the word Materials for Material

It has already been corrected.

2.2 Line 136-147: provide more details about physical-chemicals analysis

References [46], [47], [48], and [49] were added, where the details of these analytical measurements can be known.

2.3 What is the statistical analysis used in this study? And the software? Please describe them.

An L-optimal mixture response surface design was chosen in this study. Two mixtures were combined, one referring to the volumetric fractions of goat's milk whey (GMW) and the mix of strawberry and tree tomato juices (JMX), and the second referring to the mass fractions of milk kefir grains (MKG) and water kefir grains (WKG). For that reason, it is a dual-mix design. The statistical program Design - Expert version 13 was used. This program can organise the experimental design, first and later, after the experiments have been carried out, finding the statistical models that allow one or several responses to be optimised. All this is mentioned in section 2.3 (2.3. Combined dual-mix L-optimal response surface design), on lines 111-127.

3. Results 

3.1 Line 195-197: The words that indicate the figures need to be patronized in accordance with the legend of the figure. For example: If the term 1a stays in the text, the same term need describe in the legend of the figures.

Already corrected in the text. In addition, to facilitate the understanding of Figure 1, where an attempt is made to summarise the quality of the statistical models, comments were introduced for each of the "rows" and "columns". The "rows" correspond to the LAB, yeast, and acceptability models. At the same time, the "columns" refer to the normalised graph, the t-studentized distribution of the residuals and the correspondence between the values predicted by the model and the experimental values achieved.

4. Conclusions

4.5 Conclusions it's extensive. This topic can be reduced, focusing mainly on results obtained, and adding information about DPPH results. 

Considering that some of the comments have already been addressed in other parts of the manuscript, the 'Conclusions' were shortened according to your suggestion.

Part of the suggested expansion of the DPPH results was added to the end of the 'Results' section on lines 319-328.

5. And please, respond to the following question: what is the real relevance of DPPH analysis for this study? In all MS this theme is not well explored.

The presence of antioxidant compounds in these beverages and other bioactive compounds may suggest their nutraceutical or functional character. In the present study, we only tried to determine the different levels of antioxidant capacity of the candidate beverages that had reached the best behaviours of the desirability function ('OPT1', 'OPT2', and 'OPT3') and compared them with those of the "raw materials" used in its preparation (GMW and JMX). In all cases, there was an increase in the antioxidant capacity of the beverages, showing that, in some way, the kefir grains contribute to increasing these bioactive compounds.

Subsequent studies must be carried out to find out, however, how the water kefir grains in the absence of GMW and using only the mixture of strawberry and tree tomato juices, and the possible presence of proteins, sugars and vitamins of vegetable origin, were capable of being transformed until reaching similar levels of antioxidant capacity than those found in goat's milk whey.

Reviewer 3 Report

1.In part 2.2, what volume of mixed juice can be obtained from 3590 g of fresh ripe fruit (2000 g of tree-tomato + 1590 g of strawberry) mixed with 1.2L sterilized water?

2.In this section 2.4, the method for the determination of lactic acid bacteria and yeast is missingï¼›

3.The conclusion is too long, it is recommended to refine and compress to a paragraph

Author Response

Thank you very much for reviewing our manuscript.

Next, I will answer each of the remarks and answer your questions and suggestions.

1.In part 2.2, what volume of mixed juice can be obtained from 3590 g of fresh ripe fruit (2000 g of tree-tomato + 1590 g of strawberry) mixed with 1.2L sterilized water?

After consulting with the other authors, there was a mistake in writing the procedure here. No water was added during the preparation of the fruit pulp mix. Instead, 3590 g of strawberry + 2000 g of tree tomato in a conventional blender were added and blended for 30-40 s until the mixture was homogeneous. A total volume of approximately 1.2 L was reached at the end.

This has already been corrected in the new version of the manuscript.

2.In this section 2.4, the method for the determination of lactic acid bacteria and yeast is missingï¼›

It is true, you are right. This part has already been added to the manuscript. It was inserted between lines 147-152.

3.The conclusion is too long, it is recommended to refine and compress to a paragraph

This has already been fixed. In the new manuscript version, the conclusions were cut and left in a single paragraph.

Round 2

Reviewer 2 Report

Dear authors,

Thanks for the responses and improvement of the manuscript.